# Effects of Thermal Treatment on the Mechanical Properties of Bamboo Fiber Bundles

**DOI:** 10.3390/ma16031239

**Published:** 2023-01-31

**Authors:** Jie Cui, Daixin Fu, Lin Mi, Lang Li, Yongjie Liu, Chong Wang, Chao He, Hong Zhang, Yao Chen, Qingyuan Wang

**Affiliations:** 1Institute of New Energy and Low-Carbon Technology, Sichuan University, Chengdu 610065, China; 2Failure Mechanics and Engineering Disaster Prevention, Key Laboratory of Sichuan Province, Sichuan University, Chengdu 610065, China; 3Key Laboratory of Deep Earth Science and Engineering, Ministry of Education, Sichuan University, Chengdu 610065, China

**Keywords:** moso bamboo, thermal treatment, fiber bundles, facture behavior, tensile properties

## Abstract

Bamboo is known as a typical kind of functional gradient natural composite. In this paper, fiber bundles were extracted manually from various parts of the stem in the radial direction, namely the outer, middle, and inner parts. After heat treatment, the mechanical properties of the fiber bundles were studied, including the tensile strength, elastic modulus, and fracture modes. The micromechanical properties of the fiber cell walls were also analyzed. The results showed that the mean tensile strength of the bamboo fiber bundles decreased from 423.29 to 191.61 MPa and the modulus of elasticity increased from 21.29 GPa to 27.43 GPa with the increase in temperature. The elastic modulus and hardness of the fiber cell walls showed a positive correlation with temperature, with the modulus of elasticity and the hardness increasing from 15.96 to 18.70 GPa and 0.36 to 0.47 GPa, respectively. From the outside to the inside of the bamboo stems, the tensile strength and elastic modulus showed a slight decrease. The fracture behavior of the fiber bundles near the outside approximates ductile fracture, while that of the bundles near to the inside tend to be a brittle fracture. The fracture surfaces of the bamboo bundles and the single fibers became smoother after heat treatment. The results show that bamboo fiber bundles distributed near the outside are most suitable for industrial development under heat treatment at 180 °C. Therefore, this study can provide a reasonable scientific basis for the selective utilization, functional optimization, and bionic utilization of bamboo materials, which has very important theoretical and practical significance.

## 1. Introduction

In the past several decades, fiber-reinforced composites have been widely used in various construction fields, transportation fields, furniture items, and many other fields [1]. Glass and carbon fibers are popular due to their good mechanical and thermal properties [2]. However, these fibers require complex preparation processes and high costs, which can cause serious environmental problems especially when the products are scrapped [3]. Therefore, natural fibers are considered to be a more sustainable alternative for structural composites [4]. As a kind of natural plant fiber, bamboo fiber is an ideal fiber reinforcement material due to its easy availability and low cost, as well as its high strength and aspect ratio [5]. Because of its great significance in environmental protection and resource conservation, bamboo fiber has gradually become a current research hotspot [6].

From a biomaterial perspective, the gradient composite structure of bamboo is a masterpiece of nature from the micro- to macro-scale, as shown in Figure 1. On the cell scale, each sublayer of the cell wall can be regarded as a nanofiber-reinforced composite, with cellulose microfibrils acting as the reinforcement and flexible hemicellulose and lignin acting as the matrix; on the tissue scale, bamboo fibers and parenchymal cells with multiple wall layers are formed by laminar composites between cell wall sublayers [7]; and on the macro-scale, bamboo fibers are bonded to form fiber sheaths by middle lamellar and together with vessels and other conducting tissues to form vascular bundles, which are distributed in flexible parenchyma tissues in forming two-phase composites, as shown in Figure 1b During the reinforcing phase, vascular bundles determine almost all of the mechanical properties, and their distribution density increases continuously from the inside to the outside in the radial direction, which makes bamboo material have the characteristic of being a mechanical gradient functional material [8].

For the development of natural bamboo fiber composites, currently, it is crucial to study the macroscopic and microscopic mechanical properties of bamboo fibers as well as their chemical properties. Currently, bamboo fibers are extracted by chemical and steam explosion methods [9], as shown in Table 1. Chen et al. [10] tested the tensile strength and elastic modulus of single bamboo fibers extracted by a chemical method and the results were 1.77 GPa and 26.85 GPa, respectively, while the tensile strength and elastic modulus of the fiber bundles were 0.61 GPa and 23.56 GPa, respectively. The tensile strength and elastic modulus of the bamboo fibers extracted by the steam explosion method were 0.62 to 0.86 GPa and 35.45 GPa, respectively [11]. However, the tissue structure and chemical composition of bamboo fibers are inevitably affected by either chemical or steam explosion treatments [2], which affects their mechanical properties. Shao et al. [12] prepared tensile specimens of bamboo vascular bundles without physical and chemical treatments and measured their tensile strength and elastic modulus to be 0.48 GPa and 33.9 GPa, respectively, but the paper and glue deformed slightly during the stretching process which affected the overall experimental results. Meanwhile, Li et al. [13] investigated the effect of radial distribution on vascular bundles extracted by alkali treatment, and the results showed that the longitudinal tensile strength and elastic modulus of vascular bundles increased linearly from the inside to the outside of the bamboo stem.

In practical applications, heat treatment is generally used to improve dimensional stability, mold resistance, and durability [14]. The heat treatment of bamboo can be divided into steam heat treatment, oil heat treatment, and air or inert gas treatment [15]. In the current study, the main focus is on the effects of heat treatment on chemical composition and less so on the macroscopic and microscopic mechanical properties of bamboo. Azadeh et al. [16] tested the mechanical properties of bamboo in six temperature ranges from 100 °C to 225 °C and found no significant changes in mechanical properties below 150 °C, while dimensional stability and mildew resistance were improved. The relative crystallinity of cellulose increases as well as the mechanical properties after steam heat treatment at 180 °C. For example, the hardness of the cell wall increased from 0.69 GPa to 0.84 GPa [17] and the elastic modulus increased from 17.3 GPa to 21.2 GPa [18].

In summary, bamboo fibers have a wide range of applications and good prospects for development. In addition, heat treatment technology can improve the mechanical properties of bamboo fiber cell walls while effectively improving dimensional stability. According to previous studies, the traditional fiber bundles tensile test is performed with paper as the clamping section, which will produce relative slip. Moreover, data on the mechanical properties of bamboo fiber bundles after heat treatment are very scarce due to increased experimental difficulty as the bamboo fiber bundles become brittle after heat treatment. In addition, the effect of heat treatment on the mechanical properties of fiber bundles in terms of radial distribution needs to be explored further.

In this study, natural bamboo fiber bundles were extracted manually, and the conventional tensile test method was optimized. The macroscopic and microscopic mechanical properties, chemical composition, fracture mode, and the change law of the desired properties in the radial direction of bamboo fiber bundles after heat treatment were investigated by nanoindentation, FTIR, SEM, and other characterization methods. This study improves the accuracy of the fiber tensile tests and helps to select the appropriate bamboo fiber part and heat treatment temperature in the production of bamboo fibers, which has some referential significance for optimizing the production of bamboo-fiber-reinforced composites and the future development of advanced composites.

## 2. Materials and Methods

### 2.1. Specimen Preparation

3-year-old moso bamboo from Sichuan Province China was adopted as the raw material. Bamboo stems at a height of about 2.9 m to 3.2 m above the ground were selected, as shown in Figure 2a. After harvesting, the stem was dried naturally for 90 days under laboratory conditions (25–33 °C and 60–70% relative humidity), and the final moisture content was measured to be 8%. Then, the stem was split into 10 mm-wide strips and divided into three equal parts in the radial direction (Figure 2b). According to the location of the parts, they were named as the outer layer, middle layer, and inner layer. To separate the fiber bundles from the stem, the strips were squeezed repeatedly with a roller and then soaked in distilled water for one week; then, the fiber bundles in the bamboo strips were manually peeled out.

To measure the hardness and elastic modulus of the fibers in different parts, the test area of the nano-indentation specimen was 10 × 5 mm (10 mm was the radial direction), and the thickness of the bamboo block was 3 mm.

### 2.2. Heat Treatment

The chemical composition of bamboo remains unchanged when the temperature is lower than 160 °C and the highest application temperature of heat treatment at present is 220 °C [15]. Therefore, 160 °C, 180 °C, 200 °C, and 220 °C were chosen as the heat treatment temperatures. Before heat treatment, the above specimens were heated to 65 °C in the electric thermostatic drying oven for 24 h. Then, the dry specimens were subjected to the target treatment temperature for one hour. After heat treatment was completed, the temperature was naturally cooled down to the ambient temperature in the oven.

### 2.3. Tensile Test

Fiber bundles with a length of 60 mm and uniform thickness were selected, and the cross-section of fiber bundles was photographed with an optical microscope (Olympus GX53, Olympus Corporation, Tokyo, Japan). Then, the cross-sectional area of each fiber bundle was analyzed with OLYCA m3 software, as shown in Figure 2e. The two ends of the fiber bundle were glued to 30 mm × 15 mm × 1 mm acrylic plates with epoxy resin glue (3M DP6310, Minnesota Mining and Machinery Manufacturing Company, Saint Paul, USA) and the bonding depth was 10 mm. Therefore, the standard distance section of the tensile specimen was 40 mm. The other two acrylic plates were glued to both ends with acrylic plate special glue, and the fiber bundles were sandwiched in the two acrylic plates to minimize slippage during the stretching process. A piece of paper in the size of 70 mm × 30 mm was glued between the two acrylic plates to protect the fiber bundle from being broken and then cut before loading, as shown in Figure 2f. The test equipment was a fatigue tester (MMT-250, Shimadzu, Kyoto, Japan) from the brand Shimadzu with a loading capacity of 250 N. The loading speed of the specimens was 0.12 mm/min. Six specimens were tested for each group of specimens. According to the statistics, this method could make 96.7% of the fiber bundles break in the effective area, which greatly improved the experimental success rate.

### 2.4. Nanoindentation Test

A nanoindentation test (KLA iNano, KLA Corporation, Chandler, AZ, USA) performed by a Berlovich indenter was used to study the hardness and elastic modulus of the bamboo fibers. Before testing, the surface of the specimen was finely polished. In all of the tests, the maximum load was 500 μN, and the rate of loading and unloading was 50 nm/s. The load was held for 6 s at the maximum load, which was used to eliminate the effect of creep. Forces and displacements were recorded simultaneously to obtain load–depth curves. Parameters such as elastic modulus and hardness were calculated according to the Olive–Pharr method by the following equation [19]:(1)$$H=\frac{P}{A}$$
(2)$$A=24.56h_{c}^{2}$$
(3)$$h_{c}=h-\varepsilon \frac{P}{S}$$
(4)$$E_{r}=\frac{\sqrt{\pi }}{2\beta }\times \frac{S}{\sqrt{A}}$$
(5)$$\frac{1}{E_{r}}=\frac{1-v^{2}}{E}+\frac{1-v_{i}^{2}}{E_{i}}$$
where P is the peak indentation load; A is the projected area of the indentation at the maximum indentation depth; $h_{c}$ is the contact depth derived from the synthetic load–displacement curve; h is the maximum penetration depth; ε is the geometric constant of the indenter. For conical indenters, the empirical value of ε is 0.75; S = d*P*⁄d*h* (stiffness) is the slope of the upper half of the unloading phase of the load–displacement curve; $E_{r}$ is the fold modulus; commercial nanoindenters are usually taken as β = 1.034 for Berkovich indenters; the elastic modulus $E_{i}$ and Poisson’s ratio $v_{i}$ of diamond indenters are 1440 GPa and 0.07, respectively [20]. To calculate E (the elastic modulus of the sample), it is necessary to know the Poisson’s ratio v of the sample in the direction of the test. In general, the elastic modulus is not sensitive to the value of the Poisson’s ratio, and the cell walls of moso bamboo are much softer than diamonds, which leads to it taking no account of the effect of the Poisson’s ratio v of the sample [21]. Each probe indentation point was on a single fiber, and 30 valid data were selected for each of the outer, middle, and inner areas.

### 2.5. SEM Test

A scanning electron microscope (Hitachi SU3500, Tokyo, Japan) was used to examine the fracture morphology of the fiber bundle specimens under an accelerating voltage of 15 kV. The samples were gold coated and fixed on the carrier table with conductive adhesive.

### 2.6. FTIR Test

Fiber bundles after heat treatment at different temperatures were collected and an FTIR analysis was conducted to analyze the chemical composition change. Firstly, the bamboo was grinded to powder with a grinding bowl; then, the powder was dealt with by a tablet machine to obtain tablet samples for FTIR analysis. The analysis was conducted with FTIR from the brand Invenio at Sichuan University. The scanning speed was set to 16 times per minute and the resolution was 4 cm^−1^. The spectra within the wavenumber of 400~4000 cm^−1^ were recorded.

## 3. Results and Discussions

### 3.1. Tensile Strength and Elastic Modulus

The stress–strain curves, the tensile strength and elastic modulus of the bamboo fiber bundles at different heat treatment temperatures are shown in Figure 3 and Table 2. The curves at each temperature showed nearly perfect straightness characteristics with no slip or yielding bending segments. Therefore, the fracture of moso bamboo fiber bundles was a brittle fracture. The mean tensile strength of the unheated specimens was 423.29 MPa and the strength decreased gradually with the increase in temperature, and finally, it decreased to 191.61 MPa at 220 °C. However, the elastic modulus increased with the increasing treatment temperatures, which gradually increased from 21.29 GPa for the untreated specimen to 27.43 GPa for the treated specimen at 220 °C. In addition, the fastest increase in the elastic modulus was observed after 200 °C. In contrast, the two mechanical property differences of moso bamboo in the radial direction (Table 2) showed an increasing trend of both tensile strength and elastic modulus from the inside to the outside.

The above tensile experiments were performed for the macroscopic mechanical properties of bamboo fiber bundles. To further analyze the differences and changes in the internal mechanical properties of bamboo materials, the elastic modulus and hardness of individual fiber cell walls in different parts before and after heat treatment were measured by nanoindentation tests, and the results are shown in Figure 4.The elastic modulus and hardness of bamboo fibers were positively correlated with the temperature, and the elastic modulus and hardness of the fiber cell walls increased by 17.17% and 30.56% from the initial 15.96 GPa and 0.36 GPa values at 220 °C, respectively. In addition, the growth rate of both mechanical properties after 200 °C and the changes in the radial direction were also consistent with the results of tensile experiments. The increase in the elastic modulus of microscopic fiber cell walls from the inside to the outside of the bamboo diameter is essentially responsible for the increase in the elastic modulus of macroscopic fiber bundles. Moreover, the intercellular layer, which has weak adhesive strength, did not affect the fiber bundles composed of multiple fiber cells due to temperature changes, and the individual fibers still showed a consistent pattern of change with the fiber bundles.

According to previous studies, the tensile strength of untreated moso bamboo fiber bundles ranges from 290–610 MPa and the elastic modulus ranges from 15.85–28.5 GPa [12,22,23], so the test results of this experiment were relatively accurate. The reasons for the discrepancies were not only due to basic factors such as the age, height, and origin of the bamboo, but also the differences in extraction methods and testing methods. Traditional chemical and physical extraction methods lead to changes in fiber cell structure, and the specimen used by this paper as the gripping end inevitably led to slippage during the tensile process. The mean elastic modulus and hardness of untreated bamboo cell walls ranges from 10.4–17.3 GPa and 0.22–0.49 GPa, respectively [17,18,24], and the results of this experiment were also within reasonable limits. The elastic modulus values obtained from the nanoindentation were smaller than those measured in the tensile experiments. On the one hand, the nanoindentation test only measured the elastic modulus of the cell wall of the fiber part; on the other hand, the surface of bamboo is relatively fragile, and the cell wall would be damaged inevitably in the process of sample preparation, resulting in a small measurement value. It has also been shown that the widely used Oliver–Pharr analysis method greatly underestimates the longitudinal elastic modulus of anisotropic plant cell walls [25], but this discrepancy is not a major concern of this work.

In addition, the mechanical properties and change law of bamboo fiber bundles after heat treatment were studied for the first time, while ensuring the accuracy of the experimental results and increasing experimental precision. This study provides favorable reference data for the later development of bamboo-fiber-reinforced fiber composites and the application of heat treatment technology.

### 3.2. FTIR Analysis

In previous studies, it has been shown that cellulose is the determining factor affecting the longitudinal mechanical properties of fibers [26], while lignin contributes to the elastic modulus [27]. The study of the changes in chemical functional groups and the compositional structure of bamboo fiber bundles during different heat treatments was of great importance for the analysis of their mechanical properties.

Hemicellulose was easier to decompose than any other chemical component due to its amorphous structure [18], and the non-conjugated C=O stretching vibration of hemicellulose xylan was represented at a peak of 1737 cm^−1^ [28], as shown in Figure 5. The intensity of the characteristic peak decreased with increasing temperature, indicating the degradation of hemicellulose. Cellulose has characteristic peaks at 3340 cm^−1^ (O-H stretching vibration) and 900 cm^−1^ (C-H bending vibration). The intensity of the peaks at 3340 cm^−1^ and 900 cm^−1^ decreased slightly but not significantly with increasing temperature, showing that the degradation degree of cellulose was small, which was due to the good stability of cellulose [29]. The main reason for the degradation of cellulose was that a large number of hydroxyl groups in the amorphous region of bamboo cellulose underwent an oxidation reaction under the combined action of high-temperature heat treatment and oxygen-containing conditions, which leads to the polymerization of the free hydroxyl groups and the formation of aldehyde, ketone, or carboxyl groups [15]. Due to the relatively stable benzene ring structure of lignin [30], the degradation of cellulose and hemicellulose after heat treatment and lignin condensation reactions thus led to an increase in the relative lignin content [31]. The intensity of the absorption bands of lignin at 1512 cm^−1^ (C-H aromatic skeleton vibration [32]) and 1242 cm^−1^ (C-O stretching vibration [33]) remained essentially unchanged until 200 °C and only started to decrease at 220 °C.

As indicated by the FTIR analysis, both cellulose and hemicellulose underwent some degree of degradation with the increase in heat treatment temperature, which led to a significant decrease in the tensile strength of the fiber bundles. The increase in the relative lignin content was chemically responsible for the increase in the elastic modulus of the fiber bundles. Due to the decomposition of hemicellulose during heat treatment, the elastic modulus of the cell wall increased, which also resulted in a reduction of hydrogen bonding on the fiber surface, thus enhancing the micromechanical properties of the bamboo cell wall [18].

### 3.3. Analysis on Fracture Surface

Tensile damage of bamboo fiber bundles begins with relative sliding between cell walls. On a molecular scale, the abundance of hydroxyl groups in the cellulose molecular chains [34] leads to relative sliding between the cell walls of bamboo fibers involving abundant hydrogen bond formation, breakage, and recombination. As the load continues to increase and the stress reaches a level sufficient to break the covalent bonds, the internal covalent bonds between C-C and C-O receive disruption and larger molecules become smaller ones. After that, the stress continues to increase, with the cellulose crystalline region disrupted and the main skeleton disintegrating at the molecular level, which eventually leads to cell wall destruction and single fiber breakage. To better study the damage modes of different specimens, SEM was used to record the fracture of each specimen.

From the fracture surfaces of the specimens (Figure 6), before 200 °C, the fracture of the specimens near the outside was serrated, accompanied by a large amount of fiber extraction, and the fracture behavior could be regarded as a ductile fracture, while the fracture of the specimens near the inside was relatively neat and resembled a brittle fracture. After 200 °C, the fracture of the fiber bundles in different areas showed a flatter shape, and only a small number of fibers were extracted from the fracture near the outside of the tensile specimen. For the same part of the bamboo, the higher the heat treatment temperature, the smoother the fracture surface of bamboo bundles.

The fracture modes of moso bamboo single fibers can be classified into two modes: multilevel delamination fracture (Figure 7a—outer, Figure 7b—outer and middle) and near-flat fracture (Figure 7c–e). Multilevel delamination fracture is due to the different deformation capacities of thick and thin layers in the fiber cell wall containing different microfibril angles (MFAs) and lignin content [35,36]. As the tensile force increased, the thick and thin layer interfaces continuously separated, tore, and even delaminated, resulting in the formation of a multilevel delamination fracture mode. This fracture pattern generally occurred in the outer and middle extracted fiber bundles before 180 °C. With the increase in the heat treatment temperature and the degradation of internal chemical composition and small molecules, the bonding of the cell wall layers gradually loosened along with the appearance of the pores. Both the extracted fibers (Figure 7c) and the flat fracture surface (Figure 7d,e) showed the mode of near-flat fracture.

Temperature causes some degree of damage to the fibers, usually in the form of increased gaps between cell wall layers and the loss of bulk tissue, which become more pronounced as the temperature increases. In addition, it is most noticeable at 220 °C and not significant before 220 °C, which is also consistent with previous studies [33]. Furthermore, it is known that the changes in the cell structure caused by heat treatment are the physical cause of the decrease in tensile strength. Interestingly, under the same heat treatment temperature, the fracture morphology of single fibers at different sites also differed slightly, with the fracture near the inner side being rougher than the outer side, indicating a greater influence by temperature. This phenomenon becomes more pronounced at higher temperatures, especially at 220 °C (Figure 8e). Differences in fiber fracture of different parts of bamboo after high temperature heat treatment explain the phenomenon that the tensile strength of the outer fiber bundles is higher than that of the inner ones at the same temperature. In addition, the schematic fracture mode of the bamboo bundles and fibers is shown in Figure 8.

The temperature could cause some degree of damage to the fibers, usually in the form of an increased number of pores between the cell wall layers and the shedding of bulk tissue, which became more pronounced as the temperature increased. Therefore, the changes in cell structure that were caused were the physical cause of the decrease in tensile strength after heat treatment. It is interesting that the fracture morphology of single fibers in different areas also differed slightly under the same heat treatment temperature, with the fracture surface near the inside of the bamboo stem being rougher than the outside, showing that the temperature had a greater influence on the inner fibers of the bamboo stem. This phenomenon becomes more pronounced at higher temperatures, especially at 220 °C (Figure 7e). The structural differences in the different areas with temperature explained the phenomenon that the tensile strength of the outer fiber bundles is greater than that of the inner ones at the same temperature.

It is worth noting that the stress–strain curve of the fiber bundles showed the characteristics of brittle fracture when the heat treatment temperature was below 200 °C, while the outer and partially inner fiber bundle fractures exhibited the mode of ductile fracture. Tensile damage of bamboo fibers bundles begins with relative sliding between cell walls. As can be seen in Figure 3a–d, the stress–strain curve of the fiber bundles contains very small stepwise jitters. When the curve jitters, some of the fibers have relative slippage and internal damage is formed. At this time, the overall specimen was not damaged, and brittle fracture occurred to the fibers when the ultimate strength was reached. The fracture mode was characterized by a ductile fracture due to the slippage between the fiber cells during the stretching process.

More pores and a loss of bulk tissue appeared in the fiber fracture with the increase in the heat treatment temperature, indicating a decrease of fiber strength. These pores were squeezed by the two wall layers and played a buffering role throughout the tensile process. The highest number of pores between the cell walls was found at 220 °C. During the tensile process, the slip between the fibers had not yet occurred when the pores were not reduced completely, while the fibers were not able to withstand the load and were destroyed. The fracture was relatively flat because there was no slip, which results in a brittle fracture at high temperatures. In addition, the characteristics of the stress–strain curve correspond to the microstructure.

From the previous analysis, higher heat treatment temperatures lead to the degradation of cellulose and hemicellulose thus decreasing the tensile properties of bamboo fibers and increasing the elastic modulus and hardness. At the same time, the fiber bundles distributed on the outer side have better mechanical properties. In practice, the heat treatment temperature can be increased if a higher elastic modulus and stiffness are required. On the contrary, the heat treatment temperature can be reduced if a higher tensile strength is required. In general, a heat treatment temperature of 180 °C is a good choice.

## 4. Conclusions

This study aimed to investigate tensile properties, elastic modulus, and tissue structure as well as the variation law in the radial distribution of artificially extracted natural fiber bundles by heat treatment techniques. The following conclusions can be drawn from the results:

FTIR shows that an increase in temperature leads to degradation of chemical composition, with a decrease in the relative content of cellulose and hemicellulose and an increase in the relative content of lignin. The tensile strength of the bamboo fiber bundles gradually decreased with the increasing temperature from 432.29 MPa at the beginning to 191.61 MPa at 220 °C, which was mainly due to the decrease in the relative content of cellulose and hemicellulose. The elastic modulus of bamboo fiber bundles showed an opposite trend, increasing from 21.29 GPa to 27.43 GPa with the increasing temperature, which was mainly due to the decrease in relative lignin content. In the radial direction, the elastic modulus and tensile strength of the fiber bundles near the outside of the bamboo were greater than those near the inside. In addition, the elastic modulus of the cell wall showed the same law as the elastic modulus of the macroscopic fiber bundles.

The fracture behavior of the bamboo fiber bundles near the outside resembled ductile fracture, while the fracture behavior of the bamboo fiber bundles near the inside approximated brittle fracture. The fracture mode of the fibers also changed from multilevel delamination fracture to near-flat fracture when the temperature increased, and the differences in the fiber cell wall structure in the radial direction became more obvious. The change in intrinsic structural organization was the main reason for the decrease in tensile strength. Remarkably, the fracture of fiber bundles is a brittle fracture, while it produces a fracture in ductile fracture mode at low temperatures because of cell wall slippage. Therefore, bamboo fiber bundles distributed near the outside are most suitable for industrial development under heat treatment at 180 °C.

The bond strength between the cell walls and the significant increase in elastic modulus at 200 °C–220 °C still need to be further explored in future research work. Quantitative analysis of chemical composition and physical modeling of cell wall fracture will be the focus of future research work. This study is of great importance for the selection of heat treatment temperatures, the utilization of bamboo materials, and the development of green and sustainable infrastructure construction materials.

## Figures and Tables

**Figure 1 materials-16-01239-f001:**
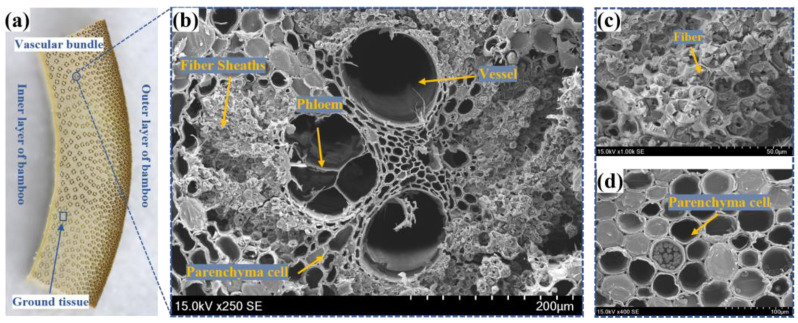
The macroscopic and microscopic structure of moso bamboo: (**a**) radial section of bamboo; (**b**) microstructure from the cross-section; (**c**) fiber sheaths observed from the cross-section; (**d**) parenchymal cells observed from the cross section.

**Figure 2 materials-16-01239-f002:**
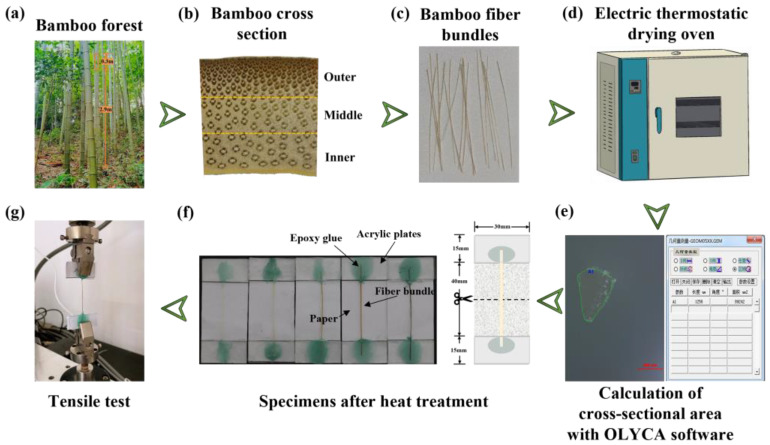
Processing steps and experimental process.

**Figure 3 materials-16-01239-f003:**
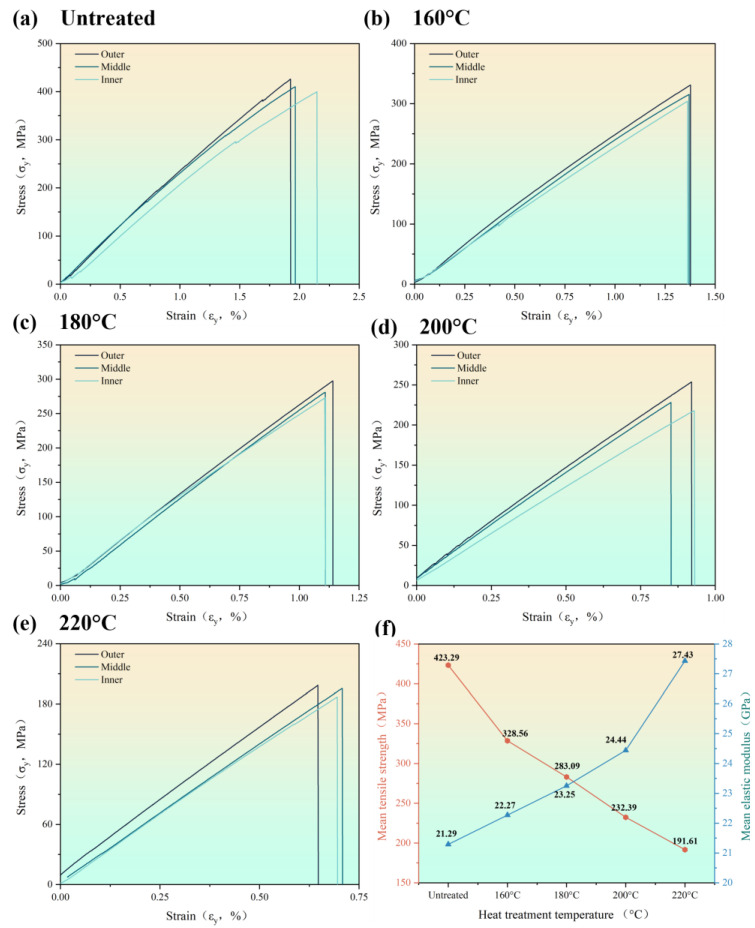
Tensile behavior of the untreated specimen: (**a**) the stress–strain curve of the untreated specimen; (**b**) the stress–strain curve of the specimen at 160 °C; (**c**) the stress–strain curve of the specimen at 180 °C; (**d**) the stress–strain curve of the specimen at 200 °C; (**e**) the stress–strain curve of the specimen at 220 °C; (**f**) the mean tensile strength and elastic modulus of bamboo bundles.

**Figure 4 materials-16-01239-f004:**
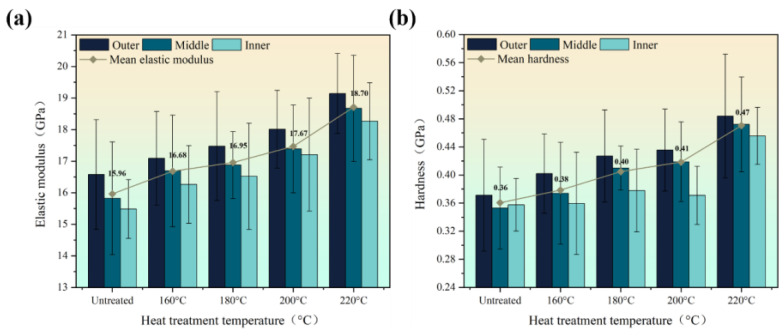
The micromechanical properties of the fiber cell walls in different areas in the radial direction: (**a**) elastic modulus and (**b**) hardness.

**Figure 5 materials-16-01239-f005:**
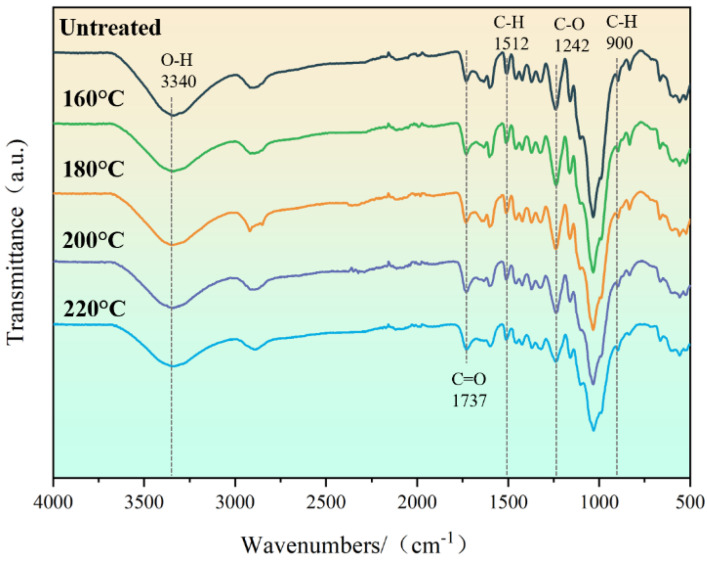
FTIR analysis of untreated and heat-treated specimens.

**Figure 6 materials-16-01239-f006:**
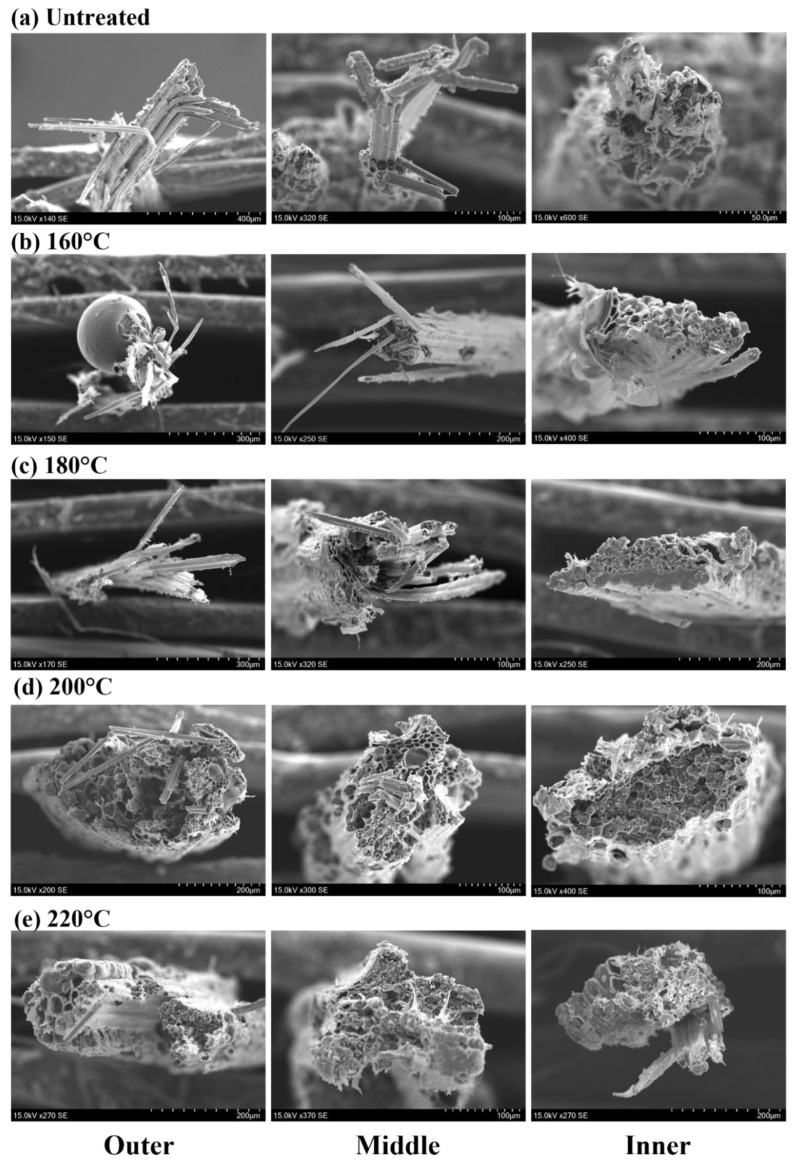
Overview of fracture surfaces in different areas in the radial direction with different temperatures.

**Figure 7 materials-16-01239-f007:**
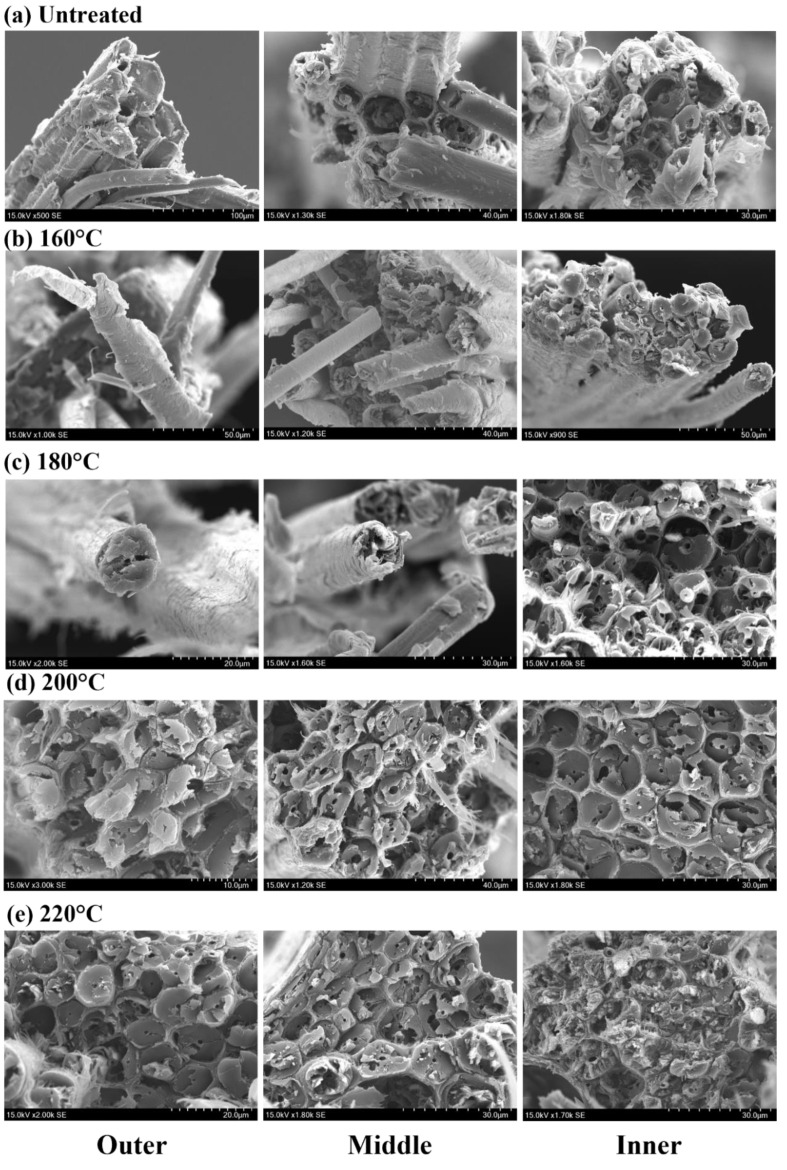
Overview of fracture surfaces of fibers in different areas in the radial direction with different temperatures.

**Figure 8 materials-16-01239-f008:**
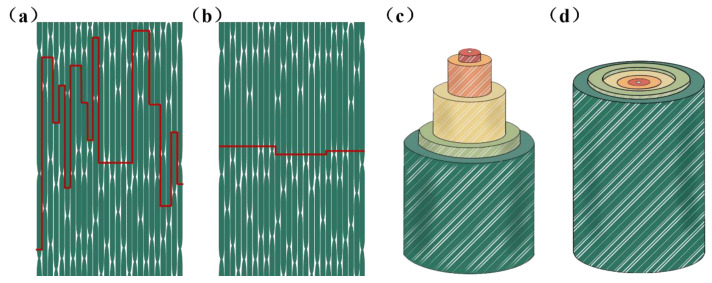
Schematic fracture mode of bamboo bundles and fibers: (**a**) ductile fracture mode of bamboo bundles; (**b**) brittle fracture mode of bamboo bundles; (**c**) multilevel delamination fracture; (**d**) near-flat fracture.

**Table 1 materials-16-01239-t001:** Extraction methods and mechanical properties of bamboo fibers and fiber bundles.

Materials	Extraction Method	Tensile Strength (GPa)	Elastic Modulus (GPa)	References
Bamboo fibers	Chemical method	1.77	26.85	[10]
	Steam explosion method	0.62–0.86	35.45	[11]
Bamboo fiber bundles	Chemical method	0.61	23.56	[10]
	Manual extraction method	0.482	33.9	[12]

**Table 2 materials-16-01239-t002:** Tensile strength and elastic modulus at different heat temperatures.

Temperature (°C)	Area in Radial Section	Mean Tensile Strength of Fiber Bundles (MPa)	Mean Elastic Modulus of Fiber Bundles (GPa)	Mean Elastic Modulus of Fiber Cell Wall (GPa)	Mean Hardness of Fiber Cell Wall (GPa)
Untreated	Outer	442.36	22.75	16.58	0.37
	Middle	421.93	20.87	15.82	0.35
	Inner	400.46	20.26	15.48	0.36
160 °C	Outer	322.49	23.70	17.09	0.40
	Middle	314.41	22.26	16.69	0.37
	Inner	310.69	21.01	16.26	0.36
180 °C	Outer	295.94	24.05	17.48	0.43
	Middle	280.58	23.88	16.88	0.41
	Inner	272.24	22.44	16.52	0.38
200 °C	Outer	246.38	25.33	18.02	0.44
	Middle	228.53	24.95	17.39	0.42
	Inner	217.90	23.31	17.21	0.37
220 °C	Outer	193.17	28.00	19.14	0.48
	Middle	190.43	27.36	18.67	0.47
	Inner	185.68	26.41	18.27	0.46

## Data Availability

Not applicable.

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
