# Peer review of "Effects of Thermal Treatment on the Mechanical Properties of Bamboo Fiber Bundles"

_materials, 2023, doi:10.3390/ma16031239_

Round 1

Reviewer 1 Report

Very well written Manuscript performed experiment very thoroughly described, and all data and assumptions for the experiment are given. The whole work ends with well-written conclusions.

My comments:

Line 98: "3-year-old moso bamboo from Sichuan province China was adopted as the raw ma- 98
terial."
Does age play any role in the final properties?

Line 118: "drying oven for 24 hours" Is there time to reduce this time?

Line 193: This figure is the same as in the line 201?

Τou should take some electron microscopy pictures to show how temperature affects the microstructure of the material, because in the photos of figure 8 you do not see any change. Is this what you want?

The manuscript contains important scientific results.

The manuscript corresponds to the Materials.

Its subject matter is compatible with the Journal.

Author Response

Response to Reviewer 1 Comments

Very well written Manuscript performed experiment very thoroughly described, and all data and assumptions for the experiment are given. The whole work ends with well-written conclusions.

Response: Thank you for your interests in our paper, and the comments you give are very helpful to improve this paper. We have exactly followed your suggestions and revised the manuscript thoroughly according to your comments. Please see the details below and feel free to let us know if there is any further suggestion on our manuscript.

Some comments:

Point 1: Line 98: "3-year-old moso bamboo from Sichuan province China was adopted as the raw ma- 98 terial." Does age play any role in the final properties?

Response 1: Previous study have shown that age has little effects on the tensile mechanical properties of bamboo fibers after 1 year(Dan Ren et al. [1], As a general rule, it seems that age has little effect on the tensile mechanical properties of bamboo fibers). We happen to know the exact age of the bamboo material used in this study, and we chose bamboo of that age for the rigor of the data. And the age of bamboo is quantitative in this study. Therefore, the age does not play a role in the final results.

Point 2: Line 118: "drying oven for 24 hours". Is there time to reduce this time?

Response 2: In fact, we measured the weight loss rate of bamboo fiber every hour after 15 hours. We found that the rate of mass loss was almost constant after 20 hours. Theoretically, the drying time could be reduced. In order to avoid the accident of other samples of bamboo fiber having no drying, we extended the drying time.

Point 3: Line 193: This figure is the same as in the line 201?

Response 3: Thank you for pointing it out. Figure 3 shows the stress-strain curves and the mean elastic modulus and tensile strength of the bamboo fiber bundles, and Figure 4 shows the strength and elastic modulus, which are not duplicated. However, considering your comments, we carefully checked whether the graphical data in the full text are duplicated and found that the data in Figure 4 can be fully presented by Table 1. Therefore, we deleted Figure 4 in the revised version.

Figure 3. Tensile behavior of untreated specimen: (a) the stress-strain curve of untreated specimen; (b) the stress-strain curve of specimen at 160°C; (c) the stress-strain curve of specimen at 180°C; (d) the stress-strain curve of specimen at 200°C; (e) the stress-strain curve of specimen at 220°C; (f) the mean tensile strength and elastic modulus of bamboo bundles.

Figure 4. The mechanical properties in different areas of radial direction: (a) tensile strength; (b) elastic modulus.

Table 1. Tensile strength and elastic modulus at different heat temperatures.

Temperature (°C)

Area in radial section

Mean tensile strength of fiber bundles (MPa)

Mean elastic modulus of fiber bundles (GPa)

Mean elastic modulus of fiber cell wall (GPa)

Mean hardness of fiber cell wall (GPa)

Untreated

Outer

442.36

22.75

16.58

0.37

Middle

421.93

20.87

15.82

0.35

Inner

400.46

20.26

15.48

0.36

160°C

Outer

322.49

23.70

17.09

0.40

Middle

314.41

22.26

16.69

0.37

Inner

310.69

21.01

16.26

0.36

180°C

Outer

295.94

24.05

17.48

0.43

Middle

280.58

23.88

16.88

0.41

Inner

272.24

22.44

16.52

0.38

200°C

Outer

246.38

25.33

18.02

0.44

Middle

228.53

24.95

17.39

0.42

Inner

217.90

23.31

17.21

0.37

220°C

Outer

193.17

28.00

19.14

0.48

Middle

190.43

27.36

18.67

0.47

Inner

185.68

26.41

18.27

0.46

Point 4: Τou should take some electron microscopy pictures to show how temperature affects the microstructure of the material, because in the photos of figure 8 you do not see any change. Is this what you want?

Response 4: Thanks for your comment about this. There are three main effects of temperature on fiber microstructure. The first one (yellow box), the fracture mode of fiber changes from multilevel delamination fracture to near-flat fracture with the increase of temperature. The change of the fracture mode can be clearly seen in the Figure 8. The second one( blue box) , the surfaces of fiber fracture have more bulk tissue and pores (blue arrows) with the increase of temperature. And it is most noticeable at 220°C and not significant before 220°C, which is also consistent with previous studies.( Qinming Feng et al.[2] As shown in Figure 5b and c, the number of fine pores within and between these cells increased at temperatures exceeding 200°C). The third one( red box), the inner layer has a rougher fiber surface and more pores at the same heat treatment temperature which is more obvious at higher temperature.

From Figure 8 and the above analysis, it can be seen that the microstructure affected by temperature is most obvious in the inner fibers at the highest temperature, while the outer fibers are affected by temperature mainly in the fracture mode, and the change of microstructure is not as significant as that of the inner fibers. To give the reader a better understanding of the changes in microstructure, we add arrows to the figure 8. Besides,we have added the description in line 314 as follows:

“And it is most noticeable at 220°C and not significant before 220°C, which is also consistent with previous studies[33].”

Figure 8. Overview of fracture surfaces of fibers in different areas of radial direction with different temperatures: (a) Untreated; (b)160°C;(c) 180°C; (d) 200°C; and (e) 220°C.

The references mentioned in this document are as follows:

  1. Ren, D.; Yu, Z.; Li, W.; Wang, H.; Yu, Y., The effect of ages on the tensile mechanical properties of elementary fibers extracted from two sympodial bamboo species. Industrial Crops and Products 2014, 62, 94-99.
  2. Feng, Q.; Huang, Y.; Ye, C.; Fei, B.; Yang, S., Impact of hygrothermal treatment on the physical properties and chemical composition of Moso bamboo (Phyllostachys edulis). Holzforschung 2021, 75, (7), 614-625.

Reviewer 2 Report

The topic of the article is very relevant. The article is devoted to the use of bamboo materials, which is very important. The purpose of this study was to investigate the tensile properties, modulus of elasticity of bamboo fibers. The authors established patterns of changes in the properties of bamboo fibers during heat treatment. A different nature of the fracture was established depending on temperature. The text is clear and easy to read. The conclusions are scientifically substantiated and correspond to the evidence presented. I recommend the manuscript of the article for publication

Author Response

Response to Reviewer 2 Comments

The topic of the article is very relevant. The article is devoted to the use of bamboo materials, which is very important. The purpose of this study was to investigate the tensile properties, modulus of elasticity of bamboo fibers. The authors established patterns of changes in the properties of bamboo fibers during heat treatment. A different nature of the fracture was established depending on temperature. The text is clear and easy to read. The conclusions are scientifically substantiated and correspond to the evidence presented. I recommend the manuscript of the article for publication.

Response: We are grateful for your time spent on our manusctipt and recognition of out work in this study. In order to present the best content of the article, we also carefully checked and reviewed the grammar, long sentences and experimental results.

Reviewer 3 Report

This paper studied the effect different heat treatment temperatures on the mechanical properties of the bamboo fibers. Fibers from different sections of the bamboos were extracted and studied by a serious of characterization methods. Data shows that the increasing temperature will lead to a decrease in elastic modulus and increase in strength. Heat treatment can provide a smoother fracture surface. This reports can potentially be helpful for the future development of bamboo fiber products.

This article is in general well structured and the design of experiment is reasonable. The language and presentation of the content needs to be polished. Some typos and grammar mistakes are found in multiple sections. The expression can be improved to make the presentation more clear and logically smooth. Additional test is necessary to better support the point. This article would be more convincing if the author can dive deeper into the data and provide more quantitative test data and be consistent in the discussion section.

Specific comment:

1.       The background paragraph starting from line 57 compares the mechanical properties of bamboo fibers extracted by different method, it would be more straightforward and clearer to have a table here listing all the data from literature. It would better to use uniform unit for comparison (i.e. GPa). Same for the next paragraph, it would make people need to revisit the previous part to get an idea of the comparison.

2.       The background section is not strong enough to prove the significance of the study, please find more information in terms of the potential application and current technical insufficiency in this part.

3.       Figure 2. Caption needs to be more descriptive. (e) what software and how you measure the cross-section area? (you mentioned in the following paragraph, it would be helpful to provide some information in the caption as well). (g) do you mean tensile test?

4.       Why do you choose this temperature range, can you provide more explanation?

5.        In the discussion, the increase of the elastic modulus is contributed by the increase of the lignin content, the FTIR result only shows the trend when the temperature is lower than 200C. There’s no direct evidence that can qualitatively or quantitatively prove this point, could you add some tests to support this?

6.       Line 304 typo 108C.

7.       TGA would be a better way to characterize material degradation, it would be helpful to add a TGA test in the method to see how the composition change with temperature.

8.       Duplicate Figure 4, please delete one.

9.       Some expressions can be improved by proofreading, for example, line 241, do you mean lignin is contributing to the elastic modulus?

10.   Based on the SEM observations, it was indicated that there are two fracture modes. Figure 3 shows that all samples have a brittle failure mode. A ductile fracture would exhibit some stepwise failure in my understanding, could you provide an explanation for this?

11.   Line 274, typo, fracture.

Author Response

Response to Reviewer 3 Comments

This paper studied the effect different heat treatment temperatures on the mechanical properties of the bamboo fibers. Fibers from different sections of the bamboos were extracted and studied by a serious of characterization methods. Data shows that the increasing temperature will lead to a decrease in elastic modulus and increase in strength. Heat treatment can provide a smoother fracture surface. This reports can potentially be helpful for the future development of bamboo fiber products.

This article is in general well structured and the design of experiment is reasonable. The language and presentation of the content needs to be polished. Some typos and grammar mistakes are found in multiple sections. The expression can be improved to make the presentation more clear and logically smooth. Additional test is necessary to better support the point. This article would be more convincing if the author can dive deeper into the data and provide more quantitative test data and be consistent in the discussion section.

Response: On behalf of all the contributing authors, I would like to express our sincere appreciations of your constructive comments concerning our article. These comments are all valuable and helpful for improving our article. According to the comments, we have made extensive modifications to our manuscript to make our results convincing. Besides, for easy recognition, these changes are also marked red in the revised manuscript.

We worked on the manuscript for a long time and the repeated addition and removal of sentences and sections obviously led to poor readability. We have now worked on both language and readability. We really hope that the flow and language level have been substantially improved.

Point 1: The background paragraph starting from line 57 compares the mechanical properties of bamboo fibers extracted by different method, it would be more straightforward and clearer to have a table here listing all the data from literature. It would better to use uniform unit for comparison (i.e. GPa). Same for the next paragraph, it would make people need to revisit the previous part to get an idea of the comparison.

Response 1: Thank you for pointing it out. We compiled the mechanical property data of the bamboo fibers and bamboo fiber bundles extracted by different extraction methods into Table 1. And, we have standardized the intensity units in the table and in the text to GPa. The addition of tables and standardized units will make it easier for readers to read and compare data.

Table 1. Extraction methods and mechanical properties of bamboo fibers and bamboo bundles.

Materials

Extraction method

Tensile strength (GPa)

Elastic modulus (GPa)

References

Bamboo fibers

Chemical method

1.77

26.85

[10]

Steam explosion method

0.62-0.86

35.45

[11]

Bamboo fiber bundles

Chemical method

0.61

23.56

[10]

Manual extraction method

0.482

33.9

[12]

Point 2: The background section is not strong enough to prove the significance of the study, please find more information in terms of the potential application and current technical insufficiency in this part.

Response 2: Thank you for your suggestion on this. We agree that the importance of this study could not be better expressed in the background section. This would have left the reader unaware of the promising applications of this study and current technical insufficiency. We have added the description to the background section as follows:

“In summary, bamboo fibers have a wide range of applications and good prospects for development. And the heat treatment technology can improve the mechanical properties of bamboo fiber cell wall while effectively improving the dimensional stability of bamboo material. According to previous studies, the traditional fiber bundles tensile test is performed with paper as the clamping section, which will produce relative slip. Moreover, data on the mechanical properties of bamboo fiber bundles after heat treatment are very scarce due to the increased experimental difficulty as the bamboo fiber bundles become brittle after heat treatment. Also, the effect of heat treatment on the mechanical properties of fiber bundles in terms of radial distribution needs to be explored further.”(line 91)

“This study improves the accuracy of fiber tensile test and helps to select the appropriate bamboo fiber part and heat treatment temperature in the production of bamboo fibers, which has some referential significance for optimizing the production of bamboo fiber reinforced composites and the future development of advanced composites.”(line 97)

Point 3: Figure 2. Caption needs to be more descriptive. (e) what software and how you measure the cross-section area? (you mentioned in the following paragraph, it would be helpful to provide some information in the caption as well). (g) do you mean tensile test?

Response 3: Thanks for your comment about this. We are very sorry for not describing the figure properly and non-professional language. In order to make Figure 2 more clear to the readers, we have made some changes on this figure. The altered parts are marked with red boxes.

Figure 2. Processing steps and experimental process: (a) bamboo forest; (b) bamboo cross section; (c) bamboo fiber bundles; (d) electric thermostatic drying oven; (e) calculation of cross-section area with OLYCA software; (f) specimens after heat treatment; (g) Fatigue tester (MMT-250) and specimens.

Point 4: Why do you choose this temperature range, can you provide more explanation?

Response 4: Thank you for pointing it out. When the heat treatment temperature is lower than 160°C, the chemical composition is basically unchanged( Zhao-Zhao Li et al.[1]. Some research showed that when the heat treatment temperature was relatively low (below 160 ◦C), the chemical composition did not change much. ). The change in mechanical strength is only due to the reduction of moisture. The maximum heat treatment temperature for current heat treatment applications is 220°C( Zhao-Zhao Li et al.[1],Table 1). It is because excessive heat treatment temperatures can lead to significant degradation of the chemical composition of the bamboo material, thus losing its use value.

To explain the reasons for selecting this range of heat treatment temperatures more clearly to the reader, the content was added in Section 2.2 as follows:

“The chemical composition of bamboo material remains unchanged when the temperature is lower than 160°C and the highest application temperature of heat treatment at present is 220°C[15]”. Line 118.

Point 5: In the discussion, the increase of the elastic modulus is contributed by the increase of the lignin content, the FTIR result only shows the trend when the temperature is lower than 200C. There’s no direct evidence that can qualitatively or quantitatively prove this point, could you add some tests to support this?

Response 5: Thank you for your suggestion on this. Cellulose and hemicellulose are degraded to a greater extent than lignin before 220°C, which leads to an increase in the relative lignin content instead. In fact, lignin is degraded, and his degradation temperature is higher than that of cellulose and hemicellulose, which is the reason why the intensity of C-H and C-O functional group peaks in Figure 5 starts to decrease only at 220°C. This result is also consistent with the results of previous studies( Xiaomeng Hao et al.[2], The decrease in intensities of absorption band at 1515 cm−1 is not obvious, which indicates the benzene ring structure of lignin is relatively stable;

Qiming Feng et al.[3] ,The intensity of the absorbance band at 1597 cm−1,…. decreased from 180 °C to 220 °C. This observation illustrates the loss of C=O groups, combined with a slight change in the intensity at 1504 cm−1 (the aromatic skeletal vibration).). Therefore, the results of the FTIR are in accordance with the universal law.

Quantitative analysis of the chemical composition of bamboo heat treatment has been previously performed in a large number of literatures( Chih-Hsuan Lee et al.[4],The results of the chemical composition analysis indicated that the content of holocelluloses, a-celluloses, and hemicelluloses declined with increasing treatment temperatures;

Qiming Feng et al.[3] , The α-cellulose content decreased by 6.3% at 220 °C compared with the control. The acid-insoluble lignin content increased with the temperature. Hemicelluloses, including pentosan, degraded continuously with the increase in temperature, and their contents decreased by 31.4 and 16.3% after treatment at 220 °C), and the results are also in agreement with the FTIR results. The aim of this study was to investigate the mechanical properties and fracture modes of bamboo fiber bundles. Therefore, we did not spend time on chemical quantitative analysis, but referred to previous research rules.

However, the significant increase in the relative content of the elastic modulus during 200°C-220°C is not based on facts, nor is it evident from the results of previous studies and FTIR. We are sorry for this uncritical conclusion. We have deleted the following two paragraphs and discussed this result in the conclusion section as a limitation of the study. Thank you again for your careful review.

“especially after 200°C, where the rate of increase in the elastic modulus was more significant.” Line 267.

“The elastic modulus showed the opposite trend and increased the fastest between 200°C and 220°C, which was mainly due to the degradation of lignin after 200°C and the increase of relative content.” lin347.

“The bond strength between the cell walls and the significant increase in elastic modulus at 200°C-220°C still need to be further explored in future research work. Quantitative analysis of chemical composition and physical modeling of cell wall fracture will be the focus of future research work.” Added in conclusion section.

Figure 6. FTIR analysis of untreated and heat-treated specimens

Point 6: Line 304 typo 108C.

Response 6: We sincerely thank the reviewer for careful reading. As suggested by the reviewer, we have corrected the “180°” into “180°C”.

Point 7: TGA would be a better way to characterize material degradation, it would be helpful to add a TGA test in the method to see how the composition change with temperature.

Response 7: Thank you for your suggestion on this. We have done the TGA test before and the experimental results are as follows:

The first weight loss (8.59%) occurred between 26–161.34 °C. This relatively small amount of weight loss was due to moisture evaporation from the fiber structure. The mass basically does not decrease much from 160°C to 200°C, but gradually starts to decrease after 220°C. The main mass loss is between 220°C and 452.08°C, during which a great deal of chemical composition was degraded. It is because a large loss of mass can cause the bamboo to lose its use value, during which a significant degradation of the chemical composition takes place. As shown in the figure, the TGA test has no research significance within the temperature range of this study. Therefore, we did not choose to include the TGA test in the article.

Point 8: Duplicate Figure 4, please delete one.

Response 8: Thank you for pointing it out. Figure 3 shows the stress-strain curves and the mean elastic modulus and tensile strength of the bamboo fiber bundles, and Figure 4 shows the strength and elastic modulus, which are not duplicated. However, considering your comments, we carefully checked whether the graphical data in the full text are duplicated and found that the data in Figure 4 can be fully presented by Table 1. Therefore, we deleted Figure 4 in the revised version.

Figure 3. Tensile behavior of untreated specimen: (a) the stress-strain curve of untreated specimen; (b) the stress-strain curve of specimen at 160°C; (c) the stress-strain curve of specimen at 180°C; (d) the stress-strain curve of specimen at 200°C; (e) the stress-strain curve of specimen at 220°C; (f) the mean tensile strength and elastic modulus of bamboo bundles.

Figure 4. The mechanical properties in different areas of radial direction: (a) tensile strength; (b) elastic modulus.

Table 1. Tensile strength and elastic modulus at different heat temperatures.

Temperature (°C)

Area in radial section

Mean tensile strength of fiber bundles (MPa)

Mean elastic modulus of fiber bundles (GPa)

Mean elastic modulus of fiber cell wall (GPa)

Mean hardness of fiber cell wall (GPa)

Untreated

Outer

442.36

22.75

16.58

0.37

Middle

421.93

20.87

15.82

0.35

Inner

400.46

20.26

15.48

0.36

160°C

Outer

322.49

23.70

17.09

0.40

Middle

314.41

22.26

16.69

0.37

Inner

310.69

21.01

16.26

0.36

180°C

Outer

295.94

24.05

17.48

0.43

Middle

280.58

23.88

16.88

0.41

Inner

272.24

22.44

16.52

0.38

200°C

Outer

246.38

25.33

18.02

0.44

Middle

228.53

24.95

17.39

0.42

Inner

217.90

23.31

17.21

0.37

220°C

Outer

193.17

28.00

19.14

0.48

Middle

190.43

27.36

18.67

0.47

Inner

185.68

26.41

18.27

0.46

Point 9: Some expressions can be improved by proofreading, for example, line 241, do you mean lignin is contributing to the elastic modulus?

Response 9: Yes, it means that lignin contributes to the elastic modulus and we revised the original article. And we tried our best to improve the manuscript and made some changes to the manuscript. These changes will not influence the content and framework of the paper. And here we did not list the changes but marked in red in the revised paper.

Point 10: Based on the SEM observations, it was indicated that there are two fracture modes. Figure 3 shows that all samples have a brittle failure mode. A ductile fracture would exhibit some stepwise failure in my understanding, could you provide an explanation for this?

Response 10: Yes, as the reviewer mentioned, the tensile fracture process of bamboo is a stepwise ductile fracture process( Huanrong Liu et al. [5] Figure 2 as follows). This is mainly due to the fact that bamboo is a multifunctional gradient composite material and the difference in mechanical properties of the fibers and parenchymal cells and the slip between the cell walls lead to a ductile fracture mode with a stepwise stress-strain curve during fracture.

Different from bamboo, this study aims to investigate the mechanical properties of bamboo fiber bundles. Without the relative slip between fibers and parenchymal cells, the fracture of bamboo fiber bundles does not produce a long stepwise facture process. But the slip between the cell walls still exists. As can be seen in Figure 3a, b, c and d , the stress-strain curve of the fiber bundles still contains very small stepwise jitters( red circles). When the curve jitters, some fibers have relative slippage and  internal damage has been formed . At this time, the specimen was not damaged and brittle fracture of fibers occured when the ultimate strength was reached. The fracture mode is characterized by ductile fracture due to the slippage between the fiber cells during the stretching process.

More pores and loss of bulk tissue appeared in the fiber fracture with the increase of heat treatment temperature, indicating a decrease in fiber mechanical properties.These pores were squeezed by the two wall layers and played a buffering role throughout the tensile process. The most pores between the cell walls were found at 220°C. During the tensile process, the slip between the fibers has not yet occurred when the pores were not reduced completely, while the fibers have not been able to withstand the load and destroyed. The fracture was relatively flat because there was no slip, which results in a brittle fracture at high temperature. And the characteristics of the stress-strain curve correspond to the microstructure.

To explain this phenomenon more clearly to the reader, the content was added in the revised version below line 323 as follows:

“It is worth noting that the stress-strain curve of the fiber bundles showed the characteristics of brittle fracture when the heat treatment temperature was below 200℃, while the outer and partially inner fiber bundles fractures exhibited the mode of ductile fracture. Tensile damage of bamboo fibers bundles begins with relative sliding between cell walls. As can be seen in Figure 3a, b, c, and d, the stress-strain curve of the fiber bundles contains very small stepwise jitters. When the curve jitters, some fibers have relative slippage and internal damage has been formed. At this time, the overall specimen was not damaged, and brittle fracture occurred to the fibers when the ultimate strength was reached. The fracture mode is characterized by a ductile fracture due to the slippage between the fiber cells during the stretching process.

More pores and loss of bulk tissue appeared in the fiber fracture with the increase of heat treatment temperature, indicating a decrease in fiber strength. These pores were squeezed by the two wall layers and played a buffering role throughout the tensile process. The most pores between the cell walls were found at 220°C. During the tensile process, the slip between the fibers has not yet occurred when the pores were not reduced completely, while the fibers have not been able to withstand the load and are destroyed. The fracture was relatively flat because there was no slip, which results in a brittle fracture at high temperatures. And the characteristics of the stress-strain curve correspond to the microstructure.”

Figure 3. Tensile behavior of untreated specimen: (a) the stress-strain curve of untreated specimen; (b) the stress-strain curve of specimen at 160°C; (c) the stress-strain curve of specimen at 180°C; (d) the stress-strain curve of specimen at 200°C; (e) the stress-strain curve of specimen at 220°C; (f) the mean tensile strength and elastic modulus of bamboo bundles.

Figure 7. Overview of fracture surfaces in different areas of radial direction with different temperatures: (a) Untreated; (b)160°C; (c) 180°C; (d) 200°C; (e) 220°C.

Point 11: Line 274, typo, fracture.

Response 11: We feel sorry for our carelessness. In our resubmitted manuscript, the typo is revised. Thanks for your correction. Besides, we have double checked all the words to make sure that there is no more mistyping in this manuscript.

The references mentioned in this document are as follows:

  1. Li, Z.-Z.; Luan, Y.; Hu, J.-B.; Fang, C.-H.; Liu, L.-T.; Ma, Y.-F.; Liu, Y.; Fei, B.-H., Bamboo heat treatments and their effects on bamboo properties. Construction and Building Materials 2022, 331.
  2. Hao, X.; Wang, Q.; Wang, Y.; Han, X.; Yuan, C.; Cao, Y.; Lou, Z.; Li, Y., The effect of oil heat treatment on biological, mechanical and physical properties of bamboo. Journal of Wood Science 2021, 67, (1).
  3. Feng, Q.; Huang, Y.; Ye, C.; Fei, B.; Yang, S., Impact of hygrothermal treatment on the physical properties and chemical composition of Moso bamboo (Phyllostachys edulis). Holzforschung 2021, 75, (7), 614-625.
  4. Lee, C.-H.; Yang, T.-H.; Cheng, Y.-W.; Lee, C.-J., Effects of thermal modification on the surface and chemical properties of moso bamboo. Construction and Building Materials 2018, 178, 59-71.
  5. Liu, H.; Jiang, Z.; Fei, B.; Hse, C.; Sun, Z., Tensile behaviour and fracture mechanism of moso bamboo (Phyllostachys pubescens). Holzforschung 2015, 69, (1), 47-52.

Reviewer 4 Report

The comments are enclosed separately.

Author Response

Response to Reviewer 4 Comments

The presented work emphasized on the influence of heat treatment on mechanical properties of bamboo fibre bundles. The presented work is interesting to the readers, however, following shortcomings are presented in the manuscript and which has to be rectified before further processing. I request mandatory revision, as listed below, please do not simply respond but revise manuscript.

Response: On behalf of all the contributing authors, I would like to express our sincere appreciations of your constructive comments concerning our article. These comments are all valuable and helpful for improving our article. According to the comments, we have made extensive modifications to our manuscript to make our results convincing. Besides, for easy recognition, these changes are also marked red in the revised manuscript.

Point 1: Similarity content is found to be 27% (Turnitin), it should be minimized to less than 20%.

Response 1: We checked the similarity content, and found that none of the repetitions were complete sentences, which were all long phrases. We considered modifying these expressions to reduce the similarity content, but were worried about causing the reading trouble to the readers. Different expressions of the same thing in different articles might not be easy for the readers to understand. We have tried our best to modify some phrases and word order while keeping the original meaning unchanged as follows:

“In this paper, fiber bundles were extracted from different parts of the stem in the radial direction, namely outer, middle, and inner.”Line 13.

“In this paper, fiber bundles were extracted manually from various parts of the stem in the radial direction, namely outer, middle, and inner.” Revision.

“After heat treatment, the tensile properties of the fiber bundles were studied, including the tensile strength, elastic modulus and the fracture modes.”Line15.

“After heat treatment, the mechanical properties of the fiber bundles were studied, including the tensile strength, elastic modulus, and fracture modes.” Revision.

And more changes can be seen in the revised version.

Point 2: The abstract should contain quantitative results such as the values of improved strength, hardness, elastic properties, etc., rather than quantitative findings.

Response 2: Thank you for your suggestion on this. We have added the values of mechanocal properties in the revised manuscript. Please see the new abstract as follows:

“Bamboo is known as a typical kind of functional gradient natural composite. In this paper, fiber bundles were extracted from different parts of the stem in the radial direction, namely outer, middle, and inner. After heat treatment, the tensile properties of the fiber bundles were studied, including the tensile strength, elastic modulus and the fracture modes. The micromechanical properties of the fiber cell wall were also analyzed. The results showed that the mean tensile strength of bamboo fiber bundles decreased from 423.29 to 191.61 MPa and the modulus of elasticity increased from 21.29 GPa to 27.43 GPa with the increase of temperature. The modulus of elasticity and hardness of the fiber cell walls showed a positive correlation with temperature, with the modulus of elasticity and the hardness increasing from 15.96 to 18.70 GPa and 0.36 to 0.47 GPa, respectively. From the outside to the inside of the bamboo stems, the tensile strength and elastic modulus showed slightly decrease. The fracture behavior of fiber bundles near the outside approximates to ductile fracture, while the fracture behavior of fiber bundles near the inside tends to be brittle fracture. After heat treatment, the fracture surfaces of bamboo bundles and the single fiber become smoother. The final reults shows that bamboo fiber bundles distributed near the outside are most suitable for industrial development at a heat treatment of 180°C. Therefore, this study can provide a reasonable scientific basis for the selective utilization, functional optimization and bionic utilization of bamboo materials, which has very important theoretical and practical significance.”

Point 3: The practical implication of the proposed work should be explained neatly in the introduction section.

Response 3: Thank you for your suggestion on this. We agree that the importance of this study could not be better expressed in the introduction section. This would have left the reader unaware of the promising applications of this study and current technical insufficiency. We have added the description to the background section as follows:

“In summary, bamboo fibers have a wide range of applications and good prospects for development. And the heat treatment technology can improve the mechanical properties of bamboo fiber cell wall while effectively improving the dimensional stability of bamboo material. According to previous studies, the traditional fiber bundles tensile test is performed with paper as the clamping section, which will produce relative slip. Moreover, data on the mechanical properties of bamboo fiber bundles after heat treatment are very scarce due to the increased experimental difficulty as the bamboo fiber bundles become brittle after heat treatment. Also, the effect of heat treatment on the mechanical properties of fiber bundles in terms of radial distribution needs to be explored further.”(line 91)

“This study improves the accuracy of fiber tensile test and helps to select the appropriate bamboo fiber part and heat treatment temperature in the production of bamboo fibers, which has some referential significance for optimizing the production of bamboo fiber reinforced composites and the future development of advanced composites.”(line 97)

Point 4: Since, the work mainly concentrated on the heat treatment. On what basis, the temperatures were selected for the treatment. Detailed justification is needed.

Response 4: Thank you for pointing it out. When the heat treatment temperature is lower than 160°C, the chemical composition is basically unchanged( Zhao-Zhao Li et al.[1]. Some research showed that when the heat treatment temperature was relatively low (below 160 ◦C), the chemical composition did not change much. ). The change in mechanical strength is only due to the reduction of moisture. The maximum heat treatment temperature for current heat treatment applications is 220°C( Zhao-Zhao Li et al.[1],Table 1). It is because excessive heat treatment temperatures can lead to significant degradation of the chemical composition of the bamboo material, thus losing its use value.

To explain the reasons for selecting this range of heat treatment temperatures more clearly to the reader, the content was added in Section 2.2 as follows:

“The chemical composition of bamboo material remains unchanged when the temperature is lower than 160°C and the highest application temperature of heat treatment at present is 220°C[15]”. Line 118.

Point 5: It is suggested to highlight the limitations of this study, suggested improvements of this work and future directions in the conclusion section. Also, the conclusion can be presented better than the present form with more findings.

Response 5: Thank you for your suggestion on this. We have added the limitations of this study, suggested improvements of this work and future directions and recapitulated the findings of this experiment in the revised manuscript. Please see the new conclusion as follows:

“This study aimed to investigate the tensile properties, elastic modulus, and tissue structure as well as variation law in the radial distribution of artificially extracted natural fiber bundles by heat treatment technics. Based on the results, the main conclusions can be drawn as follow:

FTIR shows that the increase in temperature leads to degradation of the chemical composition, with a decrease in the relative content of cellulose and hemicellulose and increase in the relative content of lignin. The tensile strength of bamboo fiber bundles gradually decreased with increasing temperature from 432.29 MPa at the beginning to 191.61 MPa at 220 °C, which was mainly due to the decrease in the relative content of cellulose and hemicellulose. The elastic modulus of bamboo fiber bundles showed an opposite trend, increasing from 21.29 GPa to 27.43 GPa with an increasing temperature, which was mainly due to the decrease in relative lignin content. In the radial direction, the elastic modulus and tensile strength of the fiber bundles near the outside of the bamboo were greater than those near the inside. Also, the elastic modulus of the cell wall showed the same law as the elastic modulus of the macroscopic fiber bundles.

The fracture behavior of bamboo fiber bundles near the outside resembled ductile fracture, while the fracture behavior of bamboo fiber bundles near the inside approximated brittle fracture. The fracture mode of the fibers also changed from multilevel delamination fracture to near-flat fracture when the temperature increased, and the differences in fiber cell wall structure in the radial direction became more obvious. The change in intrinsic structural organization was the main reason for the decrease in tensile strength. Remarkably, the fracture of fiber bundles is abrittle fracture, while it produces a fracture in ductile fracture mode at low temperatures because of cell wall slippage. Therefore, bamboo fiber bundles distributed near the outside are most suitable for industrial development at a heat treatment of 180°C.

The bond strength between the cell walls and the significant increase in elastic modulus at 200°C-220°C still need to be further explored in future research work. Quantitative analysis of chemical composition and physical modeling of cell wall fracture will be the focus of future research work. This study is of great importance for the selection of heat treatment temperature, the utilization of bamboo materials and the development of green and sustainable infrastructure construction materials.”

Point 6: Moreover, the results and discussion are not clearly dealt the outcomes of the proposed work. The authors should explicitly state the novel contribution of this work, the similarities and the differences of this work with the previous publications in this section (If possible, include some recent literatures too).

Response 6: Thank you for your suggestion on this. We have added the outcomes of the proposed work , novel contribution of this work, the similarities and the differences of this work with the previous publications of this experiment in the revised manuscript. We have added the description to the results and discussions as follows:

“In addition, the mechanical properties and change law of bamboo fiber bundles after heat treatment were studied for the first time, while ensuring the accuracy of the experimental results and increasing the experimental precision. It provides favorable reference data for the later development of bamboo fiber reinforced fiber composites and the application of heat treatment technology.” Under line 238.

“From the previous analysis, higher heat treatment temperatures lead to the degradation of cellulose and hemicellulose thus decreasing the tensile properties of bamboo fibers and increasing the elastic modulus and hardness. At the same time, the fiber bundles distributed on the outer side have better mechanical properties. In practice, the heat treatment temperature can be increased if a higher elastic modulus and stiffness are required. On the contrary, the heat treatment temperature can be reduced if the higher tensile strength is required. In general, a heat treatment temperature of 180°C is a good choice.”Under line 338.

The references mentioned in this document are as follows:

  1. Li, Z.-Z.; Luan, Y.; Hu, J.-B.; Fang, C.-H.; Liu, L.-T.; Ma, Y.-F.; Liu, Y.; Fei, B.-H., Bamboo heat treatments and their effects on bamboo properties. Construction and Building Materials 2022, 331.

Reviewer 5 Report

The authors present a study on bamboo fibers, subjected to heat treatment at various temperatures. Various studies, such as the tensile, micromechanical, and fracture properties, are performed. The literature is written very well and the experimental procedure, methods for obtaining the samples, and the equipment used are well described so that any researcher can replicate the results. The results obtained are very interesting and are a valuable addition to the literature. A minor Spelling check is required. 

FTIR and SEM are used as abbreviations and need to be expanded for their first use in the article 
Typo - Page 9 - 3.3 - Fracture is written as Fracture. 

Author Response

Response to Reviewer 5 Comments

The authors present a study on bamboo fibers, subjected to heat treatment at various temperatures. Various studies, such as the tensile, micromechanical, and fracture properties, are performed. The literature is written very well and the experimental procedure, methods for obtaining the samples, and the equipment used are well described so that any researcher can replicate the results. The results obtained are very interesting and are a valuable addition to the literature. A minor Spelling check is required. 

Response: Thank you for your interests in our paper, and the comments you give are very helpful to improve this paper. We have exactly followed your suggestions and revised the manuscript thoroughly according to your comments. Please see the details below and feel free to let us know if there is any further suggestion on our manuscript.

Point 1: FTIR and SEM are used as abbreviations and need to be expanded for their first use in the article .

Response 1: Thank you for pointing it out. FTIR and SEM have been expanded for their first use in the article. Thank you again for your careful review.

Point 2: Typo - Page 9 - 3.3 - Fracture is written as Fracture. 

Response 2: We feel sorry for our carelessness. In our resubmitted manuscript, the typo is revised. Thanks for your correction. Besides, we have double checked all the words to make sure that there is no more mistyping in this manuscript.

Round 2

Reviewer 3 Report

The updated version has responded to all the comments and significantly improved the presentation. Typo still exists, i.e. line 221 'Error, reference source not found', not sure why this happened, please proofread the article.

Author Response

Response to Reviewer 3 Comments

The updated version has responded to all the comments and significantly improved the presentation. Typo still exists, i.e. line 221 'Error, reference source not found', not sure why this happened, please proofread the article.

Response : Thank you for pointing it out. Due to the deletion of Figure 4, the serial number of the subsequent figures has changed. We have renumbered the figures in the Word version. The system generates a PDF version after submitting the Word version. We are not quite sure why the PDF version shows“ Error! Reference source not found” in line 221, but we can assure you that there is no problem in the Word version.

PDF version

Word version

In the newly submitted version of Word, we have accepted all revisions to the figure numbers. This is not the case in the PDF version generated by the system.
